New records of the archaic dolphin Agorophius (Mammalia: Cetacea) from the upper Oligocene Chandler Bridge Formation of South Carolina, USA

Boessenecker Robert W. boesseneckerrw@cofc.edu 1 2
Geisler Jonathan H. 3
1 Department of Geology and Environmental Geosciences, College of Charleston , Charleston , SC , United States of America
2 Museum of Paleontology, University of California, Berkeley , Berkeley , CA , United States of America
3 Department of Anatomy, College of Osteopathic Medicine, New York Institute of Technology , Old Westbury , NY , United States of America
Thewissen J.
Electronic publication date: 2018 Sep 28
Publication date: 2018
Volume: 6
Electronic Location ID: e5290
Received 2018 Mar 20; Accepted 2018 Jul 2
Copyright: ©2018 Boessenecker and Geisler
Copyright year: 2018
Copyright holder: Boessenecker and Geisler
License: This is an open access article distributed under the terms of the Creative Commons Attribution License, which permits unrestricted use, distribution, reproduction and adaptation in any medium and for any purpose provided that it is properly attributed. For attribution, the original author(s), title, publication source (PeerJ) and either DOI or URL of the article must be cited.
License URL: https://creativecommons.org/licenses/by/4.0/

Keywords: Odontoceti, Cetacea, Oligocene, Agorophius, Agorophiidae, Neoceti, South Carolina, Olfaction

Funding: The authors received no funding for this work.

==============================
The stem odontocete Agorophius pygmaeus (Ashley Formation, lower Oligocene, South Carolina; 29.0–26.57 Ma) has been a critical point of comparison for studies of early neocete evolution owing to its early discovery as well as its transitional anatomy relative to archaeocete whales and modern odontocetes. Some time during the late nineteenth century the holotype skull went missing and has never been relocated; supplementary reference specimens have since been recently referred to the species from the Ashley Formation and the overlying Chandler Bridge Formation (upper Oligocene; 24.7–23.5). New crania referable to Agorophius sp. are identifiable to the genus based on several features of the intertemporal region. Furthermore, all published specimens from the Chandler Bridge Formation consistently share larger absolute size and a proportionally shorter exposure of the parietal in the skull roof than specimens from the Ashley Formation (including the holotype). Furthermore, these specimens include well-preserved ethmoid labyrinths and cribriform plates, indicating that Agorophius primitively retained a strong olfactory sense. These new crania suggest that at least two species of Agorophius are present in the Oligocene of South Carolina, revealing a somewhat more complicated taxonomic perspective.

Introduction

The holotype skull of Agorophius pygmaeus consists of a partial cranium and tooth (MCZ 8761) collected in January 1847 by F.S. Holmes (Curator, College of Charleston Museum) and L.R. Gibbes (Professor, College of Charleston) from exposures of the Ashley Formation at Greer’s Landing near Middleton Place Plantation west of Charleston, South Carolina. Owing to its transitional morphology between Eocene archaeocetes and modern odontocetes, early studies often referred to Agorophius as a key comparative taxon (Whitmore & Sanders, 1977:308–310). Poor taxonomic practices led to the treatment of the Agorophiidae as a wastebasket taxon for many disparate early odontocete species now placed in other such families as the Xenorophidae, although that practice has now largely been abandoned (Whitmore & Sanders, 1977; Fordyce, 1981; Godfrey et al., 2016). The holotype originally belonged to F.S. Holmes, and after some initial study by Louis Agassiz in 1848-1850, the specimen was loaned to Joseph Leidy in 1869; by 1907, the specimen was realized to be lost (see Fordyce, 1981; Godfrey et al., 2016). In 1980, the holotype tooth was rediscovered and described by R.E. Fordyce (1981), though the skull is still missing. Fordyce (1981) published a supplementary description of the skull based on high quality illustrations.

Recently reported specimens of Agorophius have provided new reference specimens for the taxon as well as preserve aspects of the anatomy not represented in the now lost holotype cranium. These include ChM PV 4256, a partial skull and mandible with isolated teeth and associated postcrania from the Chandler Bridge Formation, and SC 2015.51.1, a partial skull from the Ashley Formation. The former specimen was identified and coded as Agorophius pygmaeus by Geisler & AES (2003) and figured by Godfrey et al. (2016); it remains undescribed. SC 2015.51.1 was described and referred to Agorophius pygmaeus by Godfrey et al. (2016). Godfrey et al. (2016) conducted a specimen-level cladistic analysis of Agorophius, and recovered a monophyletic Agorophius including these specimens and the holotype.

This study reports newly recovered specimens of Agorophius from the Chandler Bridge Formation include two incomplete skulls. These skulls raise questions about the identification of previously referred specimens of Agorophius and provide new information on ontogenetic variation and sensory anatomy in Agorophius.

Materials and Methods

Anatomical terminology follows Mead & Fordyce (2009) and Godfrey, Geisler & Fitzgerald (2013). All photographs were taken with a Canon EOS Rebel XS and 80 mm zoom lens.

Geologic background

The new crania reported in this study (CCNHM 1921, 1922) were collected by Steven Hildenbrandt in May 2017 from a construction site (now developed) in the Coosaw Preserve subdivision of North Charleston, Charleston County, South Carolina (Figs. 1A–1B). Specimens were collected in situ from the Chandler Bridge Formation exposed in an excavated pond. Other specimens collected from the same pond include a possible squalodelphinid dolphin skull and skeleton (CCNHM 2115) and a partial shell of the sea turtle Carolinachelys wilsoni (CCNHM 1903). Sediment associated with these skeletons consists of lightly consolidated, tan, silty, fine-very fine friable sandstone, consistent with bed 2 of the Chandler Bridge Formation (Fig. 1C). Based on the lithology of this bed, its relationships to other facies, and micro- and macrofossils, Katuna, Geisler & Colquhoun (1997) interpreted this bed forming in a bay or estuarine environment. At the Coosaw Preserve this bed was below the water level of the pond, thus observations of thickness, geometry, sedimentary structures, and ichnology were not possible.

Figure 1 Geographic and geologic context of Agorophius sp. specimens in South Carolina.

(A) Generalized geologic map of Oligocene strata in the Charleston area with star indicating the locality that produced CCNHM 1921 and 1922 (B) map modified from Weems & Lewis (2002) and Boessenecker, Ahmed & Geisler (2017a); (C) idealized stratigraphic section of the Chandler Bridge Formation, with silhouette indicating bed that produced the new specimens, modified from Katuna, Geisler & Colquhoun (1997).

The Chandler Bridge Formation is 24.7–23.5 Ma in age based on 87Sr/86Sr ratios from the same unit and from the overlying Edisto Formation reported by Weems et al. (2016; see also Boessenecker & Fordyce, 2017: 456–458). The shark assemblage reported from the Chandler Bridge Formation, most likely from the basal marine facies, is suggestive of inner to middle shelf depths and temperatures ranging from 20–25°C (Cicimurri & Knight, 2009). The billfish Aglyptorhynhcus similarly indicates middle shelf depths and temperatures of 20–24°C (Purdy et al., 2001; Fierstine & Weems, 2009). Overall deposition of the Chandler Bridge Formation reflects a regressive parasequence deposited over the Ashley Formation without a transgressive unit (Katuna, Geisler & Colquhoun, 1997), similar to other sequences further north on the Atlantic Coastal Plain with thin, entirely regressive parasequences are deposited under conditions of low subsidence rate (Kidwell, 1993). The unconformity between the Ashley Formation and the Chandler Bridge Formation represents approximately 2–3 million years (Weems et al., 2016). A rich assemblage of marine vertebrates is now known from the Chandler Bridge Formation including sharks, bony fish, an estuarine crocodile, freshwater and sea turtles, marine birds, cetaceans, and sirenians (Whitmore & Sanders, 1977; Sanders, Weems & Lemon, 1982; Erickson & Sawyer, 1996; Domning, 1997; Sanders & Barnes, 2002; Cicimurri & Knight, 2009; Fierstine & Weems, 2009; Geisler, Colbert & Carew, 2014; Weems & Knight, 2013; Velez-Juarbe & Domning, 2014; Weems & Sanders, 2014; Ksepka, 2014; Churchill et al., 2016; Godfrey et al., 2016; Weems & Brown, 2017).

Systematic paleontology

Mammalia Linnaeus, 1758	
Cetacea Brisson, 1762	
Odontoceti Flower, 1867	
Agorophiidae Abel, 1914	
Agorophius Cope, 1895	
Agorophius sp.	
Figs. 2–4	

Material: CCNHM 1921, partial skull including dorsal parts of parietal, supraoccipital, posterior part of frontal and incomplete right supraorbital process, partial ascending process of right maxilla, posterior tip of right premaxilla, and ethmoid; CCNHM 1922, partial skull including dorsal parts of parietal, supraoccipital, frontal, ethmoid, and a fragment of the ascending process of the right maxilla. Both specimens collected on June 19, 2017 by Steven Hildenbrandt.

Figure 2 Skulls of Agorophius. sp. (CCNHM 1921, 1922) from the Chandler Bridge Formation (A–E).

CCNHM 1922 in dorsal view (A) CCNHM 1921 in dorsal view (B) line drawings of CCNHM 1922 and 1921 in dorsal view (C, D) CCNHM 1922 in ventral view (E) CCNHM 1921 in ventral view. Abbreviations: fr, frontal; m, maxilla; n, nasal; p, parietal; px, premaxilla; so, supraoccipital.

Locality: Coosaw Preserve, exposure of Chandler Bridge Formation in excavated pond (now developed) on construction site, North Charleston, South Carolina, USA (Fig. 1). Detailed locality information on file at CCNHM, available on request to qualified researchers.

Age: early Late Oligocene (Chattian), dated to 24.7–23.5 based on 87Sr/86Sr ratios reported from mollusks by Weems et al. (2016).

Identification: CCNHM 1921 and 1922 preserve two features unique to Agorophius amongst all odontocetes with an intertemporal constriction (see diagnosis in (Godfrey et al., 2016):157): (1) parietals exposed dorsally at the intertemporal constriction without a sagittal crest, and (2) region around the vertex including the frontal, parietal, ascending maxilla, and anterior occipital shield with a flat, planar margin in lateral view. A third feature is preserved only in CCNHM 1922: a deeply incised premaxillary cleft in the posterior nasal process of the premaxilla. Owing to their occurrence in a geochronologically younger stratum and an anteroposteriorly shorter exposure of parietal at the intertemporal constriction than the holotype and referred specimen SC 2015.51.1 from the Ashley Formation, these specimens are identified only to the genus level (see ‘Discussion’).

Description

Premaxilla

In CCNHM 1922, the nasal process of the premaxilla (Figs. 2A–2B) is present is and divided into a posterolateral plate and a posteromedial splint. Although the posterolateral plate is not preserved, its existence can be inferred from the sutural surface on the maxilla. The posteromedial splint is a narrow, vertical, and posteriorly tapering wedge between the frontal (medially) and the ascending process of the maxilla (laterally). A deeply incised premaxillary cleft is vertical, parasagittally oriented, and presumably separated the posteromedial splint (preserved) from the posterolateral plate (inferred). Part of the right nasal may be present but owing to damage it is unclear if it what is observed is actually the medial edge of the premaxilla. Though broken, the premaxilla transversely widens anteriorly based on the shape of the articular “buttress” for the premaxilla on the maxilla.

Maxilla

The ascending process of the maxilla is transversely expanded, dorsoventrally thin and sheet-like, and extends posteriorly to the level of the occipital shield apex (Figs. 2A–2B). Medially, the ascending process of the maxilla forms a vertical plate that abuts the posteromedial splint of the premaxilla. This plate transitions anteriorly into a slightly raised platform that underlies the premaxilla adjacent to the nares. A single, large and posteriorly directed dorsal infraorbital foramina is present lateral to the posterior termination of the nasal, about 1 cm lateral to the premaxilla. A vertical fissure separates the maxilla from the premaxilla.

Frontal

The median frontal suture is unfused in CCNHM 1922 and obliterated in CCNHM 1921 (Figs. 2A–2B). The frontoparietal suture is transverse and W-shaped with a median point, bilateral anterior embayments, and an anterolaterally extending portion that contributes a small part to the supraorbital process. The latter part of the parietal resembles the morphology in Ashleycetus planicapitis, although in that taxon this part of the parietal expands further anterolaterally (Sanders & Geisler, 2015). Dorsally the frontal has a median, rectangular plateau outlined laterally by the premaxilla, the parietal posteriorly, and the nasals anteriorly. The frontal is anteriorly pointed with an anterolaterally facing articular surface for the nasal. In CCNHM 1921, the frontal is more acutely pointed and the rectangular median plateau is proportionally narrower. Given the higher degree of suture closure and inferred older ontogenetic stage of CCNHM 1921, this suggests that the premaxilla and maxilla grow dorsomedially, at the expense of the frontal, with age. Obviously this suggestion will be speculative until larger sample sizes are available. Dorsally, the frontal is very porous in comparison to the occipital shield and the ascending process of the maxilla.

A thin band of frontal was probably exposed posterior to the ascending process of the maxilla in CCNHM 1921, but damage to the maxilla obscures the extent. The dorsal surface of the supraorbital process of the frontal bears numerous radial, anteroposteriorly directed ridges and grooves for the ascending process of the maxilla.

The frontal forms most of the ventral surface of the compound supraorbital process (Figs. 2E–2F). The posterior edge is formed by the anterolateral wing of the parietal. Posterior to the frontal groove the frontal is smooth and shallowly concave. This part bears a series of faint, radially oriented (i.e., lateral to posterolaterally directed) vascular sulci that emanate from the medial end of the orbit. The frontal groove is partially preserved in CCNHM 1921, widens from 3 mm to about 9–10 mm laterally, is deeply concave in cross-section, and oriented 35–40°from the sagittal plane.

Ethmoid

Internally, the frontal articulates with the ethmoid and forms part of the ethmoid labyrinth (Figs. 2E–2F). The ethmoid labyrinth is best seen on the left side of CCNHM 1922 (Figs. 2E, 4A). There are two longitudinal ridges that separate 3 fossae; from dorsal to ventral they are: a larger smooth-floored fossa for the dorsal nasal meatus, a middle and undulatory fossa with various foramina, and a smooth but small fossa. Although not preserved, it is likely that these ridges became more prominent anteriorly, thus forming more distinct ethmoturbinals and more discreet nasal meatuses. The ethmofrontal suture is evident dorsally, and fused internally within the olfactory peduncle canal. The canal is transversely narrow, anteroposteriorly long, and widens anteriorly. Anteriorly the roof of the canal bears a few foramina with associated longitudinal sulci. There are no apparent exits for these foramina on the dorsal side of the skull, and we suspect that they drained the diploe of the frontal bone, as is seen in a protocetid and the basal mysticete Coronodon havensteini (Godfrey, Geisler & Fitzgerald, 2013, Geisler et al., 2017). Posteriorly the olfactory peduncle canal narrows and has a deeply incised dorsal fissure (Figs. 2E, 4B).

The ethmoid also has a prominent median portion. In anterior view it present a broken, dorsoventral partition that separates the right and left ethmoidal labyrinths. In CCNHM 1921, only a median break remains to indicate where this median projection of ethmoid connected to the rest of the skull (Figs. 2E–2F); however, the dorsal part of this median projection is preserved in CCNHM 1922 (Fig. 4B). On the right side, this median portion of ethmoid bears a transverse, broken perforate shelf that forms part of the cribriform plate (Fig. 4A). Interestingly, it does not appear to completely bridge the canal, unlike the morphology seen in archaeocetes, extant mysticetes (Godfrey, Geisler & Fitzgerald, 2013) and Squalodon (Godfrey, 2013). The plate bears a median, posteriorly directed conical projection (=crista galli).

The olfactory peduncle canal is anteroposteriorly longer and transversely narrower in CCNHM 1921, where it is developed as a narrow fissure (Figs. 2F, 4). Longitudinal sulci are present in the anterior half of the canal. The ethmoturbinal recess has some poorly preserved turbinates, but the labyrinth and recess are small and separated by a low ridge immediately adjacent to the dorsal meatus. The preserved part of the meatus is a dorsally situated, lozenge-shaped, smooth cavity.

Ventrally, the frontal descends to terminate at large, articular surfaces for the alisphenoid; these facets are rectangular, face anteroventrally, and exhibit anteroposteriorly directed ridges and furrows forming a mortised frontal-sphenoid suture. Posterior to this in CCNHM 1921, the frontals form the anterodorsal margin of the endocranial surface with a median fissure-like opening for the olfactory nerve canal.

Parietal

Dorsally each parietal forms a triangular exposure, and medially they are connected by an anteriorly bowed, narrow band wrapping around the apex of the occipital shield (Figs. 2A–2B). A distinct crease occurs between the supraoccipital and the parietal. The frontoparietal suture in CCNHM 1921 is open and anastomosing; the suture is more highly mortised and mostly closed in CCNHM 1922 (Figs. 2A–2B). The parietal at the midline is pinched between the supraoccipital and frontal, and this exposure is reduced in CCNHM 1922, which we attribute to ontogenetic development of cranial telescoping.

The lateral edge of the parietal is concave in dorsal view, forming the medial margin of the temporal fossa; the anterolateral wing of the parietal extends along the posteromedial part of the compound supraorbital process of the frontal and buttresses it ventrally. The intertemporal region of the skull is dorsally flat with a nearly contiguous surface with the frontal (Figs. 3A–3B). The intertemporal portions of the frontal and parietal are very cancellous in CCNHM 1922 (Fig. 2A). In CCNHM 1921, three small foramina are present on either side of the parietal (Fig. 2B), though less clearly developed in CCNHM 1922. The lateral side of the braincase is broadly concave and formed by the parietal, which posterodorsally forms the lateral part of the nuchal crest and overhangs the temporal fossa. The frontoparietal suture is sigmoidal and anterodorsally oriented in lateral view.

Figure 3 Skulls of Agorophius sp. in lateral view.

CCNHM 1922 (A), and CCNHM 1921 (B); size comparison of Agorophius spp. Crania in dorsal view (C); scatterplot of parietal width and parietal length at the midline, in millimeters (D). Closed circles denote specimens from the Ashley Formation and open circles denote specimens from the Chandler Bridge Formation.

Supraoccipital

The occipital shield bears a triangular apex; the apex is somewhat more truncated in CCNHM 1921 than in CCNHM1922. A nuchal tuberosity is present and developed as a flat, diamond-shaped plateau at the apex. The occipital shield is transversely concave, and at the midline is inclined posteroventrally from the plane formed by the frontoparietal ‘table’ approximately 48°in CCNHM 1922 and 59°in CCNHM 1921.

Discussion

Olfactory anatomy

Several recent studies have investigated the anatomy of the ethmoid labyrinth and cribriform plate in extinct and extant cetaceans, and surprisingly found that modern mysticetes maintain a well-developed olfactory system complete with cribriform plate that differs little from archaeocete ancestors (Thewissen et al., 2011; Godfrey, Geisler & Fitzgerald, 2013). Modern odontocetes lack such structures and as a result have probably lost their olfactory sense (Edinger, 1955; Berta, Ekdale & Cranford, 2014). These anatomical changes are mirrored by molecular changes; cetaceans have a much higher proportion of olfactory receptors that are pseudogenes than many other mammals (Kishida et al., 2007; McGowen, Clark & Gatesy, 2008), and inactivation of individual genes is much more prevalent among odontocetes than in mysticetes (Springer & Gatesy, 2017). Unlike extant odontocetes, archaic Miocene odontocetes like Squalodon retain a cribriform plate and labyrinth (Godfrey, 2013). Obfuscating these matters somewhat is the apparent lack of a cribriform plate in xenorophids such as Albertocetus meffordorum and Inermorostrum xenops (Boessenecker, Ahmed & Geisler, 2017; Boessenecker et al., 2017). It remains unclear whether the condition in xenorophids detected thus far reflects true anatomy or postmortem taphonomic damage.

Figure 4 Skull of Agorophius. sp. (CCNHM 1922) in anterior view.

Skulls of Agorophius sp. reported herein shed new light on these matters, as Agorophius is typically recovered on the odontocete stem in an intermediate position between the basal position of xenorophids and later diverging Squalodon (Godfrey et al., 2016; Boessenecker et al., 2017). Both specimens (CCNHM 1921, 1922) preserve an ethmoid labyrinth and CCNHM 1922 preserves an incomplete cribriform plate (Figs. 2E, 4). It is unclear whether or not the cribriform plate completely or incompletely bridged the junction between the ethmoid labyrinth and olfactory canal, owing to damage. Regardless, some degree of a cribriform plate is present in Agorophius as well as later diverging stem odontocetes like Squalodon (Godfrey, 2013). Clarification regarding the condition in xenorophids is needed, as the apparent absence would imply two distinct losses of the cribriform plate, one due to lack of ossification (i.e., xenorophids) and the other due to closure of the olfactory foramina (crown Odontoceti).

Cranial variation and stratigraphic origin of Agorophius specimens

These newly described partial crania are incomplete but represent the first formally described specimens of Agorophius from the Chandler Bridge Formation. CCNHM 1921 and 1922 are identifiable as Agorophius but notably differ from the holotype of Agorophius pygmaeus and SC 2015.51.1 in exhibiting an anteroposteriorly shorter exposure of the parietal in dorsal view. In fact, in all specimens of Agorophius from the Chandler Bridge Formation (ChM PV4256, CCNHM 1921, CCNHM 1922), the length of the parietal at the midline is only 11.9–14.7% of the minimum parietal breadth at the intertemporal constriction. In crania from the Ashley Formation (holotype, SC 2015.51.1), the parietal is somewhat anteroposteriorly longer and constitutes 22–46.4% of the intertemporal width (Table 1). Another feature differentiating ChM PV4256 from Agorophius pygmaeus is the presence of a triangular rather than parabolic or convex apex of the occipital shield (Godfrey et al., 2016: 165). A triangular apex is also present in CCNHM 1921 and 1922. Given that the holotype is from the late early Oligocene Ashley Formation (Whitmore & Sanders, 1977; Godfrey et al., 2016), these consistent differences raise the possibility that specimens from the late Oligocene Chandler Bridge Formation (including ChM PV 4256) represent a geochronologically younger and as-yet undescribed species of Agorophius. For the time being, CCNHM 1921, CCNHM 1922, and ChM PV4256 are considered conspecific and identified as Agorophius sp., pending further study.

Table 1 Measurements of the intertemporal constriction of Agorophius.

Specimen	Minimum parietal width (mm)	Anteroposterior length of parietal at midline (mm)	Parietal width/parietal length	
MCZ 8761	49.5	23	46.46%	
SC 2015.51.1	54	11.9	22.04%	
ChM PV 4256	72	4.5	6.25%	
ChM PV 5852	66.1	10.5	15.89%	
CCNHM 1921	74.5	6.2	8.32%	
CCNHM 1922	85.1	7.1	8.34%	

Ontogenetic trends are apparent within Chandler Bridge Agorophius sp. CCNHM 1922 and ChM PV4256 both possess open median frontal sutures and clear frontoparietal sutures; both are mostly or totally closed in CCNHM 1921. CCNHM 1922 and ChM PV 4256 likely represent similar ontogenetic stages, though neither are juveniles owing to their large size (∼84–87% intertemporal width of CCNHM 1921; Figs. 3D, 4). Thus, we tentatively identify these specimens as subadults and CCNHM 1921 as an adult. If correct, then ontogenetic trends within this species include a steeper occipital shield, an anteroposteriorly longer median dorsal exposure of the frontal, an anteroposteriorly longer olfactory nerve canal, and blunted apex of an otherwise triangular occipital shield apex in CCNHM 1921. Curiously, the ratio of the anteroposterior length to transverse width of the parietal appears stable (anteroposterior length equals 6.25–8.34% of transverse width) throughout this crude growth series (CCNHM 1921, 1922, ChM PV 4256), suggesting minimal change in parietal exposure during postnatal ontogeny.

Further questions arise regarding the size of reported Agorophius crania. A smaller cranium from the Ashley Formation, ChM PV 5852, was tentatively considered to belong to Agorophius by Geisler & AES (2003) and identified as Agorophius sp. by Deméré, Berta & McGowen (2005). In fact, Sanders & Geisler (2015) recovered a sister-group relationship between ChM PV4256, which all studies agree can be referred to Agorophius (Geisler & AES, 2003; Godfrey et al., 2016), and ChM PV5852 in some of their phylogenetic analyses. Godfrey et al. (2016) considered this specimen as Odontoceti indet. because they recovered it as the sister taxon of Simocetus rayi. Although those authors did not report support values, the same relationship was also recovered by Lambert et al. (2017), and the branch support for this relationship was low. Thus we consider the generic attribution of ChM PV5852 a question that requires further investigation.

This cranium of ChM PV5852 measures 186.8 mm in bizygomatic width, slightly smaller than SC 2015.51.1 (206 mm) and the holotype of Agorophius pygmaeus (approximately 190 mm). ChM PV 5852, SC 2015.51.1, and the holotype cranium are all much smaller than ChM PV 4256 (248 mm). Newly referred crania CCNHM 1921 and 1922 lack squamosals but are similar to ChM PV4256 in intertemporal width. Whereas the holotype specimen exhibits a number of open and incipiently fused cranial sutures indicating immaturity, ChM PV 5852 does not and has a greater degree of suture closure than one of the skulls reported here (CCNHM 1921), suggesting the presence of a large and small morphotype of Agorophius (if ChM PV5852 is a species of Agorophius) If correct, then SC 2015.51.1 and ChM PV 5852 represent a small morph and ChM PV 4256, CCNHM 1921, and CCNHM 1922 all represent a much larger morph. However, owing to fact that the holotype is immature and smaller than adults of each morphotype, it is unclear which, if any, of these morphotypes represents Agorophius pygmaeus. Formal description of ChM PV 5852 and recently discovered Agorophius-like crania from the Ashley Formation is required to resolve these questions.

Conclusions

New specimens of Agorophius include two fragmentary crania preserving the intertemporal region and ‘vertex’ and constitute the first formally described remains of this odontocete from the Chandler Bridge Formation. These specimens share with Agorophius pygmaeus an intertemporal constriction, pointed occipital shield, and a flattened frontal-occipital ‘table’ in lateral view. The broken nature of these specimens reveals that Agorophius had a cribriform plate and large ethmoturbinal recess, suggesting a well-developed olfactory sense. The larger size and proportionally shorter exposure of parietal at the midline suggests that there are in fact two species of Agorophius, which differ in size. A better understanding of cranial ontogeny is needed to determine how these two species relate to the holotype of the genus, which is almost certainly an immature individual.

First and foremost we thank S Hildenbrandt for collecting the cetacean specimens reported within, and M Brown for donating the specimens to CCNHM. This study benefited from discussions with BL Beatty, SJ Boessenecker, M Churchill, M Gibson, and AE Sanders. Thanks to S Boessenecker (CCNHM) and M Gibson and J Peragine (ChM) for access to specimens under their care. Thanks to Stephen J Godfrey, an anonymous reviewer, and the editor JGM Thewissen for critical comments which improved the quality of this paper.

Institutional Abbreviations

CCNHM Mace Brown Museum of Natural History, College of Charleston, Charleston, South Carolina, USA

ChM Charleston Museum, Charleston, South Carolina, USA;

MCZ Museum of Comparative Zoology, Harvard University, Cambridge, Massachusetts, USA

SC South Carolina State Museum, Columbia, South Carolina, USA

Additional Information and Declarations

Competing Interests

Author Contributions

Data Availability

The authors declare there are no competing interests.

Robert W. Boessenecker conceived and designed the experiments, performed the experiments, analyzed the data, contributed reagents/materials/analysis tools, prepared figures and/or tables, authored or reviewed drafts of the paper, approved the final draft.

Jonathan H. Geisler analyzed the data, contributed reagents/materials/analysis tools, authored or reviewed drafts of the paper, approved the final draft.

The following information was supplied regarding data availability:

All newly reported specimens are curated at CCNHM, the Mace Brown Museum of Natural History, a college museum at the College of Charleston in Charleston, South Carolina. It is a public museum within a state college, and several papers cited within have already published on fossil cetaceans from CCNHM collections (Boessenecker, Ahmed & Geisler, 2017; Boessenecker et al., 2017; Geisler, Colbert & Carew, 2014).

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
