# Peer review of "New records of the archaic dolphin Agorophius (Mammalia: Cetacea) from the upper Oligocene Chandler Bridge Formation of South Carolina, USA"

_PeerJ, doi:10.7717/peerj.5290_

## Round 0.1 · original submission · Minor Revisions

This is a good paper. Please consider the issues mentioned by the reviewers, the only area that concerns me seriously is your description of olfactory anatomy. Please clarify it.

Apart from whar the reviewer says, I personally wonder if the age of the specimen may have something to do with the confusion that the reviewer reports. I have dissected some very young belugas and the cribriform plate is hardly ossified. I know, you are not working on a beluga.

·

Basic reporting

This article meets the PeerJ standards.

Experimental design

PeerJ standards met.

Validity of the findings

Findings valid and well-substantiated.

Additional comments

Nice tight article. Conclusions justified based on the limited data...we always wish for fossil skulls to be better preserved.
My only concern/caution to the authors is the significantly different slope to the occipital shield preserved in CCNHM 1921. It seems really different from all other agorophiid specimens...
Minor editorial corrections/suggestions are to be found within the body of the ms.
Best,
Stephen Godfrey

Reviewer 2 ·

Basic reporting

English language is used effectively throughout nearly all of the paper. The Introduction and Discussion are particularly clear and provide ample background information. Figures need clarification, particularly Figs. 2 and 4.

I would suggest labeling the supraorbital process, sutural surface between the maxilla and the posterolateral plate of premaxilla, premaxillary cleft, and frontal groove. These are mentioned in the text, but are not labeled on any figures.

I would suggest giving the Ethmoid its own subheading, separate from the Frontal.

Line 187: If the cribriform plate is not preserved in CCNHM 1921, why is it labeled in Fig. 2F?

Your statement in Lines 191-192 that “The ethmoturbinal recess and olfactory nerve canal are confluent via a pair of crescentic foramina” confuses me. I am not convinced of where the ethmoturbinal recess may be in CCNHM 1921 and 1922, especially since you have not labeled any ethmoturbinates. The ethmoturbinal recess and crescentic foramina are continuous in Godfrey, 2013, so I would expect for you to find something similar. The olfactory nerve canal should essentially not be continuous with the ethmoturbinal recess or crescentic foramina. The cribriform plate should block them from being continuous, except for the foramina within the cribriform plate which I would not expect to be patent in a fossil anyway. Additionally, there are two ethmoturbinal recesses (Godfrey, 2013). Your language should reflect that. You describe the cribriform plate as being lenticular. I don’t think of any part of the cribriform plate as being lenticular, except for possibly the crista galli. I’m doubtful that what you have identified as the cribriform plate is actually the cribriform plate.
In general, I am not convinced of some of your findings regarding the olfactory anatomy. Your identification of the proximal olfactory nerve tract makes sense to me, although the presence of “longitudinal sulci” (Line 196) surprise me. I would like to see an indication of where the olfactory bulbs would sit, largely because your identification of the cribriform plate seems faulty. In CCNHM 1922 and 1921, it seems the partial cribriform plate consists of a septum-like projection that closely straddles the midline. There seem to be no fossae or flat areas on it where olfactory bulbs would contact it, allowing olfactory sensory neurons to go towards areas where olfactory epithelium is located.
It appears that the olfactory nerve canal bifurcates distally. Is this true? If so, then the crescentic foramina (Fig. 4), could be were a left and right olfactory bulb would sit. Then the cribriform plate (Fig. 4) would actually be a bony septum separating the distal olfactory nerve tracts, like in Godfrey et al, 2013.

Line 228: Do you mean to switch those specimen numbers around? The nuchal crest of CCNHM 1922 seems to come to a sharper point anteriorly than in CCNHM 1921.

Line 253: I would be surprised if the “incipient” cribriform plate were to change much morphologically during life unless the individual were very young or very old. You tentatively claim CCNHM 1922 is subadult (and not juvenile) and that CCNHM 1921 is an adult (Lines 282-284). Therefore I doubt that what you have identified as a cribriform plate is an incipient cribriform plate.

Experimental design

Yes, this research clearly fills a gap in our knowledge.

Validity of the findings

Assertions made regarding olfaction are not well supported; more anatomical evidence must be clearly presented in order to convince the reader that these specimens contain as much olfactory anatomy, particularly the cribriform plate and ethmoturbinates, as claimed here.

Additional comments

Some minor errors include
Line 88: the degree sign
Line 104: misspelling
Line 136: insert the word “and”
The captions for Fig. 1 are not identical. Boessenecker et al. (2017a) isn’t cited in both.
Insert “(F)” into the caption for Fig. 2
In Fig. 4, “parietal” needs a leader line

---

## Round 0.2 · accepted · Accept

Good paper, thanks for your work!

#